# Effects of a waiting list control design on alcohol consumption among online help-seekers: protocol for a randomised controlled trial

Marcus Bendtsen ,[1] Katarina Ulfsdotter Gunnarsson,[1] Jim McCambridge[2]

¹Department of Health, Medicine and Caring Sciences, Linköping University, Linköping, Sweden
²Department of Health Sciences, University of York, York, UK

**Correspondence to**
Dr Marcus Bendtsen;
marcus.bendtsen@liu.se

## ABSTRACT

**Introduction** Sparse attention has been given to the design of control conditions in trials, despite their important role as contrasts for novel treatments, and thus as a key determinant of effect sizes. This undermines valid inferences on effect estimates in trials, which are fundamentally comparative in nature. Such challenges to understanding also makes generalisation of effect estimates complex, for example, it may not be clear to what degree real-world alternatives to the novel treatments in pragmatic trials are similar to the control conditions studied. The present study aims to estimate the effects of being allocated to a waiting list control condition.

**Methods and analysis** Individuals searching online for help to reduce their drinking will be invited to take part in a study. Individuals aged 18 years or older, who in the past month consumed six or more drinks on one occasion, or consumed 10 or more drinks the past week, will be eligible to participate. Both groups will receive identical feedback and advice on behaviour change; however, one group will be informed that they have to wait 1 month for the intervention materials. One month postrandomisation, participants will receive an email with the follow-up questionnaire measuring the primary outcomes: (1) frequency of heavy episodic drinking (defined as at study entry) in the past month; and (2) overall past week alcohol consumption. Differences between groups will be analysed using negative binomial regression models estimated using Bayesian inference. Recruitment will begin in October 2021. A Bayesian group sequential design will be employed to determine when to end enrolment (expected to be between 500 and 1500 individuals).

**Ethics and dissemination** The study was approved by the Swedish Ethical Review Authority on 2021-01-25 (Dnr 2020–06267). Findings will be disseminated in open access peer-reviewed journals no later than 2023.

**Trial registration trial** ISRCTN14959594; Pre-results.

## Strengths and limitations of this study

► Sparse attention has been given to the effects of different control conditions.
► Use of a double blind randomised controlled design allows for a valid estimate of the effects of being allocated to a waiting list control.
► The Bayesian group sequential design will allow for recruitment to not last longer than necessary, minimising the number of participants exposed to the necessary deception involved in the study.
► The trial is an online trial, which usually suffer from high rates of attrition.

contrast, typically a fraction of this attention is given to the control side. There is sparse attention to the design of control conditions in trials in the research literature.[2–6] Not only does this potentially introduce bias, and certainly complicates the interpretation of the effects estimated, it also makes generalisability problematic, with implications for research use in decision making.

To illustrate the complex nature of the generalisability issues, we borrow notation from the potential outcomes framework.[7] Let $Y_x$ represent the outcome for an individual if the individual would hypothetically receive treatment $x$. There may be a range of different treatments available, and depending on which treatment an individual actually receives, different outcomes will follow. In a two-arm RCT, the two *potential* outcomes for participants are $Y_0$ and $Y_1$, that is, the outcomes realised if they are allocated to the control condition ($x = 0$) or the intervention condition ($x = 1$). The individual level causal effect of the intervention is defined as $Y_1 - Y_0$, that is, the potential outcome under the intervention condition minus the potential outcome under the control condition (or some other mode of contrast, eg, risk or odds). Of course, when using between-subjects designs,

## INTRODUCTION

Effects estimated in randomised controlled trials (RCTs) should always be understood as contrasts between allocated conditions.[1] Since effects are contrasts, estimates are affected by either side of the comparison. While much effort is usually placed in designing and describing the novel treatment side of the

it will be impossible to know both $Y_1$ and $Y_0$ for the same individual; thus, RCTs are generally focused on the population level causal effect. With continuous outcomes, this is commonly the expected difference $E[Y_1] - E[Y_0]$ between the two conditions. We rely on the licence given to us by virtue of randomisation to claim that these are unconfounded estimates with sufficiently large numbers, protecting the internal validity of our estimates.

Ideally, $E[Y_1] - E[Y_0]$ would be an estimate that arises from an effectiveness or pragmatic trial rather than an efficacy or explanatory trial,[8 9] which can be interpreted as the effect anticipated in a real-world roll-out. $Y_1$ could be a novel potential outcome that we may offer to individuals, and it may be close to what was estimated in the trial if the study inclusion criteria were similar to routine practice, there were no marked bias and if treatment protocols are followed. However, a problem arises if $Y_0$ is not the potential outcome that would be the alternative to $Y_1$ in the real world. For instance, if $Y_0$ represents the potential outcome of being placed on a waiting list,[10] then that is not an outcome that is available to individuals in a real-world setting, except in those situations where there are waiting lists for treatment. To avoid any such problem, the control condition in the trial needs to be identical to usual care. If being placed on a waiting list in a study affects individuals in any way differently than do the real-world alternatives to $Y_1$, then the external validity of the contrast $E[Y_1] - E[Y_0]$ should be questioned, and measures need to be taken to facilitate the interpretation of the observed effect.

### Waiting list controls

Rather than withholding treatment from controls in RCTs altogether, it has been argued that it is more ethical to offer the treatment to controls after the trial period,[10] that is, placing participants on a waiting list. This type of control condition is commonly used in behavioural and rehabilitation treatments,[11] altering the nature of $Y_0$. It is not well understood how individuals react to these decisions. For example, in one smoking cessation trial, some control participants had negative feelings about having to wait and decided to quit later.[12] Researchers have long been concerned with feelings of disappointment about allocation to control conditions in trials more broadly, but we know little about how study participants subsequently act on them.[13 14] This may be especially the case in online studies when participants actively engage with recruitment procedures and may therefore have higher expectations of receiving novel treatment, rather than simply respond to in-person contacts from researchers that they have not initiated. However, the effects of being informed that one has to wait, as investigated here, may be less pronounced in online studies.[15–19] It therefore remains unclear if these reactions translate into any markedly biased effect estimates.

Indirect evidence suggests that in many circumstances we may expect participants in control conditions to change more than they ordinarily would if not involved in a study. It has for instance been found that alcohol consumption tends to decrease in control groups by approximately 20% in alcohol intervention trials.[20–22] This change is likely due to a combination of several factors, including regression to the mean,[23] assessment effects[24] and other forms of research participation effects.[25–27] A network meta-analysis of cognitive–behavioural therapy for depression[28] found in contrast that participants in waiting list control groups improved less than expected. This is in line with thinking that when patients are placed on waiting lists, they are to some extent compliant with the implicit direction; they wait and avoid instigating changes.[29]

A direct estimation of the effects of a waiting list control design on alcohol consumption was conducted among 185 hazardous or harmful drinkers recruited through newspaper advertisements in Canada.[30] A two-arm parallel groups RCT was employed, in which both groups received the same brief alcohol intervention, but one group was told that they had been allocated to the intervention arm, and the other told they were allocated to a waiting list arm. This exploratory trial found an interaction between being allocated to the waiting list group and the action subscale of the readiness to change questionnaire.[31 32] This suggested that those in the waiting list group who rated themselves above the median on the action subscale (thus being more ready for action) reported consuming more alcohol in comparison with their counterparts in the intervention group (approximately six drinks more per week and 1.3 more drinks on the occasion when they drank the most). They thus waited to reduce their drinking. While findings from the trial were mixed, it generated hypotheses important to study further. The current trial described in this protocol will address these hypotheses with a larger sample size in an online recruitment setting.

### Objectives

Using a two-arm (1:1) RCT, the objective of this study is to estimate the effects of being allocated to a waiting list control group on alcohol consumption. The primary effects being estimated are the total effect and moderation by readiness to change. We anticipate that these effects will be in the direction of higher consumption at follow-up among waiting list participants in comparison with non-waiting list participants.

### METHODS AND ANALYSIS

This protocol contains relevant items from the Standard Protocol Items: Recommendations for Interventional Trials.[33] The trial has been prospectively registered in the ISRCTN registry on 2021-02-01 and has received ethical approval by the Swedish Ethical Review Authority on 2021-01-25 (Dnr 2020–06267).

### Study setting, eligibility criteria and recruitment

Participants will be recruited through online advertisements (Google Ads), using language targeting to include

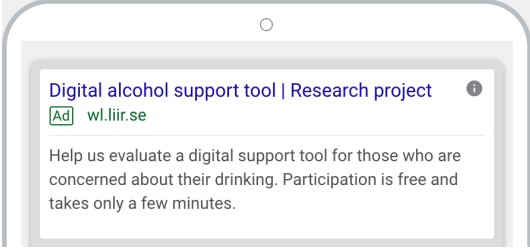

**Figure 1** Example advert shown to individuals who search online for help to drink less.

individuals using Google's services in English (including search and gmail). Keywords will be selected to target those seeking help to drink less, for example, 'How do I drink less', 'I drink too much' and 'Support for drinkers'. The adverts will be designed to target individuals who are willing to take part in a trial of a self-help alcohol intervention. An example advert is shown in figure 1, and study information presented to individuals who click on the advert can be found in online supplemental appendix A. Recruitment will begin in October 2021 and will last for no more than 24 months (October 2023, see sample size).

Individuals who consent to take part in the trial will be asked to complete a baseline questionnaire (see online supplemental appendix B). Individuals will first be asked to state their age and sex, the number of times during the past month they consumed six or more drinks on one occasion and the number of drinks consumed in the past week. Individuals will be included if they report having consumed six or more drinks on one occasion at least once in the past month or having consumed more than nine drinks in the past week. Individuals younger than 18 years old will be excluded. Those not excluded will be asked to complete the readiness to change questionnaire. After completing the questionnaire, individuals will be asked to leave an email address to which further information about study participation will be sent. The email will be sent immediately and will include a brief text explaining that a link should be clicked in the email to continue participation in the study. Individuals who click on the link in the email will be randomised to either the intervention group (IG) or the waiting list control group (WL).

### Interventions

Participants in both groups (IG and WL) will receive the same personalised feedback and advice based on the baseline questionnaire. The material, which has been developed from our previous research[24 34–40] and is based around social cognitive theories of health behaviour,[41] includes: (1) feedback on current consumption in relation to recommendations, (2) content to increase motivation and self-efficacy and (3) advice on what to do to reduce drinking and avoid environmental triggers.

Preceding the feedback and advice will be a page with study information that will be different between groups; this is the experimental contrast in the present study.

### Text to IG group

There are two groups in this study. You are in the group that receives immediate feedback and support tailored to the responses to the questionnaire you previously completed. This material has been designed for people who are concerned about their drinking. (Button: take me to the feedback).

### Text to WL group

There are two groups in this study. You are in the group that will have to wait for the feedback and support material. We will contact you again in 4 weeks and at that time you will be given the material. In the meantime, we have generated a report based on your responses to the questionnaire previously completed. (Button: take me to the feedback).

### Outcomes

**Primary**

1. Frequency of heavy episodic drinking in the past month.
2. Overall consumption the past week.

**Secondary**

3. Readiness to change.

The two primary outcomes have been selected to be similar to the outcomes used in the exploratory trial on which this trial has been modelled[30] while at the same time being part of the proposed core outcome set for brief alcohol interventions.[42 43] Frequency of heavy episodic drinking will be defined as the number of times in the past month the participant has consumed six or more drinks on one occasion. Number of drinks consumed in the past week will be assessed by asking participants to report their consumption day by day.

The secondary outcome, readiness to change, will be measured using the readiness to change questionnaire (treatment version).[31 32] While our primary use of this scale is for assessing moderation effects (using the baseline assessment), we are also interested in estimating effects of the WL condition on readiness to change directly.

### Follow-up

One month postrandomisation, participants will be sent an email including a link to the follow-up questionnaire (see online supplemental appendix B). Participants will be sent two reminders 3 days apart. If no response has been recorded 3 days after the final reminder, a fourth email will be sent with the frequency of heavy episodic drinking in the past month measure embedded in the email. Reminders will only be sent to participants who have not yet responded to the two primary alcohol consumption outcomes. Participants in the WL group will immediately after completing the outcome measure be sent an email with information about the deception involved in the study (online supplemental appendix C). A participant timeline is shown in figure 2.

| | STUDY PERIOD | | | |
|---|---|---|---|---|
| | Enrolment | Allocation | Post-allocation | Close-out |
| **TIMEPOINT** | **0** | **0** | *Access to feedback and support material* | *1-month* |
| **ENROLMENT:** | | | | |
| *Informed consent* | X | | | |
| *Eligibility screen* | X | | | |
| *Allocation* | | X | | |
| *Debriefing* | | | | X |
| **INTERVENTIONS:** | | | | |
| *Intervention group (IG)* | | | | |
| *Waiting list group (WL)* | | | | |
| **ASSESSMENTS:** | | | | |
| *Baseline questionnaire* | X | | | |
| *Follow-up questionnaire* | | | | X |
| *Accessing material* | | | X | X |
| *Time spent on material* | | | X | |

**Figure 2** Participant timeline (SPIRIT figure). SPIRIT, Standard Protocol Items: Recommendations for Interventional Trials.

## Allocation and blinding

Block randomisation will be used with randomly permuted block sizes of 2 and 4 to ensure equal size groups. Randomisation will be automatic and computerised on the backend server; thus, neither research personnel nor participants will be able to influence allocation. Participants will be blind to the true nature of the study. Research personnel will be blind to allocation.

## Statistical methods

All analyses will be conducted according to intention-to-treat principles, with all participants analysed in the groups to which they were randomised. Analyses will be done using available data with missing data imputed. Attrition analyses will explore the plausibility of the implicit missing at random (MAR) assumption underlying the analyses. Model parameters will be interpreted by inspecting marginal posterior distributions using Bayesian inference (see sample size for priors),[44] and imputation will be done using multiple imputation with chained equations. We will use R V.3.6 with the RStan library for all analyses.[45]

### Primary analyses

The analysis of primary and secondary outcomes will be conducted through regression models in which each outcome will be regressed against group allocation. One regression model per outcome will be created. Negative binomial regression will be used for both primary outcomes, and linear regression will be used for readiness to change scores (precontemplation, contemplation and action). A multinomial regression model will also be estimated for stage of change designation (ie, highest arithmetical score among the three subscales with ties preferring the stage farther along the continuum of change[31 32]). All regression models will be adjusted for age, sex and each respective outcome measured at baseline.

### Exploratory analyses

Effect modification analyses of group allocation and readiness to change scores (precontemplation, contemplation and action) on the primary outcomes will be analysed by adding interaction terms to the respective regression models. We will estimate separate models for each stage interaction and a combined model with all stages interacting with allocation using shrinkage priors.[46] We will also explore effect modification with respect to group allocation and stage of change designation on the primary outcomes.

Separate analyses of the heavy episodic drinking outcome (first primary outcome) will be conducted among those who at baseline had consumed six or more drinks on one or more occasions over the past month, thus excluding those who had no episodes of heavy drinking the month preceding enrolment. Similarly, separate analyses of the overall consumption the past week (second primary outcome) will be conducted among those who at baseline reported being at-risk with respect to weekly consumption, thus excluding those who at baseline reported having consumed nine drinks or fewer the past week.

### Mediator analyses

We will estimate the degree to which the effect of allocation on the primary outcomes is mediated through two variables: (1) accessing the intervention materials and (2) time spent on the intervention materials. We will use a causal inference framework,[47–49] using Bayesian inference to estimate the natural direct effect and natural indirect effect (as per the definitions of Pearl[49]). We will report on the posterior distributions of these two estimates, as well as the proportion of the total effect, which is accounted for by the natural indirect effect. We will also explore if readiness to change moderates any of these effects.

### Attrition analyses

The available data and imputation approach used in the primary analyses may result in biased intention-to-treat estimates if data are not MAR. We will therefore seek evidence against the MAR assumption in two attrition analyses. First, since late responders to follow-up may be more alike non-responders than early responders are, we will investigate if outcomes are systematically different given the number of reminders needed to collect follow-up data. Second, we will explore if responders and non-responders are systematically different with respect to baseline characteristics, using logistic regression models with shrinkage priors.

### Sample size

The trial will use a Bayesian group sequential design to monitor recruitment with interim analyses planned for every 25 participants completing the follow-up questionnaire.[50–52] The primary outcomes will be modelled following the analysis plan presented earlier, and each covariate representing group allocation will be assessed for evidence of effect or futility. Covariates representing the interaction between the stages of readiness for change and group allocation will also be monitored (using separate models for each stage interaction). Let $\beta_i$ represent

the covariates of interest, and $D$ the data available, then for each $\beta_i$ the target criteria will be:

▶ Effect: $p\left(\beta_i > 0 \,\middle|\, D\right) > 97.5\%$ or $p\left(\beta_i < 0 \,\middle|\, D\right) > 97.5\%$ (ie, the effect is greater (or less) than 0 with a probability greater than 97.5%).

▶ Futility: $p\left(log\left(\frac{1}{1.10}\right) < \beta_i < log\left(1.10\right) \,\middle|\, D\right) > 95.0\%$ (ie, the incidence rate ratio is greater than $1/1.10$ and less than 1.10 with a probability greater than 95%).

For the effect criterion, we will use a sceptical normal prior for covariates (mean=0 and SD=1.0), and a wider prior will be used for the futility criterion (mean=0 and SD=2.0).

The previously mentioned criteria should be viewed as targets, thus at each interim analysis, we will evaluate each criterion for each covariate and make a decision if we believe that recruitment should end. Simulations indicate that we will require a sample size in the range of 500–1500 participants. Recruitment will not exceed 24 months.

### Patient and public involvement statement

Patients were not involved in the planning of this study.

## DISCUSSION

Trials are usually advertised as important for research, but they may also be seen as an opportunity for prospective participants to get access to novel treatments. Participants have likely found the trial, and enrolled, motivated by the ways in which the trial may benefit them. However, trials do not primarily advertise the conditions that lead to $Y_0$, that is, recruitment material never says 'Do you want to be on a waiting list?'. The unintended consequences of telling participants that they will have to wait is the motivation for the present study, which is designed to uncover some of these consequences in the chosen setting.

The findings from the present study will help to understand if participants are affected by being told that they are being placed in a waiting list control group and will therefore help to guide decisions of comparator choice in future studies of brief alcohol interventions. In addition, effect estimates from the study will also support the possibility of retrospectively adjusting findings from previous trials of brief alcohol interventions to account for any bias induced by waiting list controls.[53]

### Limitations

Online studies of alcohol interventions commonly suffer from high levels of attrition, particularly at follow-up, and we expect the primary limitation of this study to be high attrition rates. To reduce attrition, we will only randomise participants after they have confirmed their email address, which helps to ensure that we have a valid email address for each randomised participant. We will use reminders over a short time span to keep participants aware that it is important that they respond and finally make it easy for them to respond by including one of the primary outcome measures in the body of the email. The Bayesian group sequential design will ensure that we collect data for as long as is necessary to fulfil the criteria (see sample size), which will ensure that we have enough samples for our analyses. However, this will not protect against differential attrition and thus does not protect against attrition bias, so we will need to study this risk carefully.

## ETHICS AND DISSEMINATION

The main ethical concern in this trial is the use of deception—required to not reveal to prospective participants that it is the effects of allocation that are being studied. Routine use of deception in research is not recommended; however, considering the widespread use of waiting list designs, understanding their effects was considered important and justified the risks induced by deception. The study was approved by the Swedish Ethical Review Authority on 2021-01-25 (Dnr 2020–06267). Data collection will begin in October 2021 and will last no longer than 24 months. The dataset will be made available to researchers on reasonable request. Findings will be published in peer-reviewed journals and presented at relevant conferences no later than 2023.

**Contributors** MB and JM defined the study objectives and designed the trial. All authors were involved in developing the intervention materials. MB and KUG are responsible for data collection and statistical analysis. All authors are involved in interpreting findings and all authors read and approved the protocol manuscript.

**Funding** The authors have not declared a specific grant for this research from any funding agency in the public, commercial or not-for-profit sectors.

**Competing interests** MB owns a private company (Alexit AB) that disseminates digital lifestyle interventions such as the one offered to participants in this trial. Alexit AB had no part in funding, planning or execution of this study.

**Patient consent for publication** Not required.

**Provenance and peer review** Not commissioned; externally peer reviewed.

**ORCID iD**
Marcus Bendtsen http://orcid.org/0000-0002-8678-1164

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
