## [Reviewer comments · BMJ Open]

ARTICLE DETAILS

TITLE (PROVISIONAL)	Effects of a waiting list control design on alcohol consumption among online help-seekers: protocol for a randomised controlled trial
AUTHORS	Bendtsen, Marcus; Ulfsdotter Gunnarsson, Katarina; McCambridge, Jim

VERSION 1 – REVIEW

REVIEWER	Kuttichira, Praveenlal Jubilee Mission Medical College and Research Institute
REVIEW RETURNED	04-Apr-2021

GENERAL COMMENTS	Title reflects the content fairly well. Abstract crisp and sharp. Introduction not exhaustive, but relevant facts narrowing down to justification of study.
---

REVIEWER	Sundström, C Karolinska Institute, Department of Clinical Neuroscience
REVIEW RETURNED	11-Apr-2021

GENERAL COMMENTS	Most randomized trials entail some form of comparison with a control group, but there is a general lack of scientific investigation into the differential effects of varying control groups, at least in online trials. This study wants to investigate the effects of the waitlist control group. More specifically, it aims to study whether receiving the intervention in question (feedback and advice on behavior change) immediately and also being informed that one receives the intervention, is more or less effective in terms of drinking reductions and stage of change (scores and designation) than being informed that one is put on a wait list after receiving the actual intervention. Thus, the experimental component in this study is the information provided with the consent when signing up for the trial. This form of research dissects aspects of research trials usually taken for granted. The current study will help build an evidence base on the impact of being put on a wait list, especially in the context of online trials. The introduction is condensed, and clearly summarizes relevant research. However, slightly more emphasis could be put on the specific context of online recruitment. A suggestion is to incorporate the text currently in the discussion into the introduction. The methods and statistical analyses all seem appropriate. In all, the protocol is very well written, interesting to read and sets out the argument clearly. I have no request for revisions, just some questions: - In the introduction, you refer to research both in support of control
---

	group improvement (in alcohol trials although you do not mention whether these control groups include wait lists, see page 6, line 3) and control group deterioration (wait list although this refers to face-to-face CBT for depression, see page 6, line 7). Do you have any hypotheses as to what you will find in the current study, i.e. which direction you expect to find effects (improvement or deterioration)? - The authors mention that participants will receive the “debriefing email” after data collection is complete. Do the authors mean after the follow-up data has been collected from the individual participant, or do you mean after the complete data collection is complete (i.e. after up to 24 months)? From figure 2 (participant timeline) it looks as if the debriefing will sent to participants immediately after the follow-up. This makes sense. However, this could be clarified in the text. - Page 6, line 58: As I understand your protocol, participants will only be included if they have either had a heavy drinking day in the past month, or have consumed more than 9 drinks in the past week. Slightly unclear about the subgroup analyses: do you mean that these analyses will be done on those who are screened at-risk for both of these outcomes at the same time (i.e. for those who report heavy drinking at least once in the past month and also report having had more than 9 drinks in the past week, in contrast to those only screened at risk from one of the two outcomes)? This could be clarified.
--	--

VERSION 1 – AUTHOR RESPONSE

REVIEWER #1

C3. Title reflects the content fairly well. Abstract crisp and sharp. Introduction not exhaustive, but relevant facts narrowing down to justification of study.

A3. Thank you for reading and commenting on the manuscript.

REVIEWER #2

C4. The introduction is condensed, and clearly summarizes relevant research. However, slightly more emphasis could be put on the specific context of online recruitment. A suggestion is to incorporate the text currently in the discussion into the introduction. The methods and statistical analyses all seem appropriate. In all, the protocol is very well written, interesting to read and sets out the argument clearly. I have no request for revisions, just some questions.

A4. Thank you. We have moved parts of the discussion regarding online recruitment to the introduction. We have added to the discussion that there opportunities to account for the potential waiting list effect in other studies which have employed them, if we have an estimate of them.

C5. In the introduction, you refer to research both in support of control group improvement (in alcohol trials although you do not mention whether these control groups include wait lists, see page 6, line 3) and control group deterioration (wait list although this refers to face-to-face CBT for depression, see page 6, line 7). Do you have any hypotheses as to what you will find in the current study, i.e. which direction you expect to find effects (improvement or deterioration)?

A5. While we are neutral to the direction of effects in our chosen priors for the Bayesian analyses, and we do not use hypothesis testing, we do anticipate that those who are told that they are on a waiting list will have a higher consumption at follow-up. We have added a sentence to the Objectives section to emphasise this: “We anticipate that these effects will be in the direction of higher consumption at follow-up among waiting list participants in comparison to non-waiting list participants.”.

C6. The authors mention that participants will receive the “debriefing email” after data collection is complete. Do the authors mean after the follow-up data has been collected from the individual participant, or do you mean after the complete data collection is complete (i.e. after up to 24 months)? From figure 2 (participant timeline) it looks as if the debriefing will be sent to participants immediately after the follow-up. This makes sense. However, this could be clarified in the text.

A6. Yes, participants in the waiting list condition will immediately after follow-up receive the debriefing email (as they are anticipating the intervention after having waited). This has been clarified in the Follow-up section: “Participants in the WL group will immediately after completing the outcome measure be sent an email with information about the deception involved in the study (Appendix C)”.

C7. Page 6, line 58: As I understand your protocol, participants will only be included if they have either had a heavy drinking day in the past month, or have consumed more than 9 drinks in the past week. Slightly unclear about the subgroup analyses: do you mean that these analyses will be done on those who are screened at-risk for both of these outcomes at the same time (i.e. for those who report heavy drinking at least once in the past month and also report having had more than 9 drinks in the past week, in contrast to those only screened at risk from one of the two outcomes)? This could be clarified.

A7. Thank you, this was quite unclear from our end. We have rephrased the entire last paragraph of the Exploratory analyses section.

VERSION 2 – REVIEW

REVIEWER	Sundström, C Karolinska Institute, Department of Clinical Neuroscience
REVIEW RETURNED	15-Jul-2021
GENERAL COMMENTS	Thank you for these changes. The manuscript is well written and my questions and comments have been appropriately addressed. I have no further comments.